**Subject Category:**
Biology (whole organism)

behaviour/ecology/statistics

fishing activities, spatial distribution, small-scale fishery, Gaussian mixture model, hidden Markov model

**Author for correspondence:**
Tania Mendo
e-mail: tm96@st-andrews.ac.uk

# Identifying fishing grounds from vessel tracks: model-based inference for small scale fisheries

Tania Mendo[1], Sophie Smout[1,2], Theoni Photopoulou[1,2,3] and Mark James[1]

[1]Scottish Oceans Institute, University of St Andrews, East Sands, Fife KY16 LB, UK
[2]School of Mathematics and Statistics, Centre for Research into Environmental and Ecological Modelling, University of St Andrews, The Observatory, Buchanan Gardens, St Andrews, Fife KY16 9LZ, UK
[3]Institute for Coastal and Marine Research, Nelson Mandela University, PO Box 77000, Nelson Mandela, Port Elizabeth 6031, South Africa

TM, 0000-0003-4397-2064; SS, 0000-0002-5125-9827; TP, 0000-0001-9616-9940; MJ, 0000-0002-7182-1725

Recent technological developments facilitate the collection of location data from fishing vessels at an increasing rate. The development of low-cost electronic systems allows tracking of small-scale fishing vessels, a sector of fishing fleets typically characterized by many, relatively small vessels. The imminent production of large spatial datasets for this previously data-poor sector creates a challenge in terms of data analysis. Several methods have been used to infer the spatial distribution of fishing activities from positional data. Here, we compare five approaches using either vessel speed, or speed and turning angle, to infer fishing activity in the Scottish inshore fleet. We assess the performance of each approach using observational records of true vessel activity. Although results are similar across methods, a trip-based Gaussian mixture model provides the best overall performance and highest computational efficiency for our use-case, allowing accurate estimation of the spatial distribution of active fishing (97% of true area captured). When vessel movement data can be validated, we recommend assessing the performance of different methods. These results illustrate the feasibility of designing a monitoring system to efficiently generate information on fishing grounds, fishing intensity, or monitoring of compliance to regulations at a nationwide scale in near-real-time.

# 1. Introduction

Sustainable management of the marine environment must take account of vulnerable and dependent species such as marine mammals, commercial resources such as fish, and human communities that depend on the marine environment for their livelihood and way of life. Management measures may involve the restriction of fishing effort to specific areas. To predict the impacts of such measures on people and marine species, it is essential to know how fishing is currently distributed in space and time, and how intense fishing is at local scales. In large scale fisheries (LSF, vessels greater than 12 m length), the introduction of vessel monitoring systems (VMS) and automated identification systems (AIS) have resulted in increased fishing vessel positional data that have allowed great progress in the identification of fishing grounds [1,2]; estimation of fishing effort [3–5], monitoring of compliance to regulations [6] and assessment of impact on marine habitats [7,8]. However, small scale fisheries (SSF) may comprise a large part of national fleets and must then also be monitored for effective fisheries management.

Knowledge of the spatial extent of fishing ground is lacking in many SSF due to the limited positional information available for this sector. However, recent technological developments of smaller, low-cost systems that can easily be fitted to smaller fishing vessels have opened up new possibilities for mapping the distribution of SSF vessels. Spatial data are now being collected in small scale fisheries operating in developing countries, for example, by using mobile applications that transmit positional data using mobile networks (https://crmg.st-andrews.ac.uk/current-projects/305-2/). These initiatives are generating timely information on the location and importance of fishing grounds that can be used for marine spatial planning in areas where almost no data were previously available to map the location of fishing activities.

In Europe, several trials are currently being conducted to assess the feasibility of using different electronic reporting systems to map the spatial footprint of SSF [9,10]. For some fleets, the reporting of positional records is already mandatory, such as the razor clam fishery in Ireland [11] or some sectors of the gillnet fishery targeting cod in Germany [12]. In the UK, there are initiatives to make the use of monitoring systems a statutory requirement in the inshore fishing fleet in England [13] and the use of appropriate vessel tracking systems has been articulated by the Scottish government in the Scottish Inshore Fisheries Strategy and a national discussion paper on the future of fishers management in Scotland [14,15].

However, monitoring on its own is not enough. Unprocessed VMS and AIS data do not indicate whether a vessel is fishing, steaming, or in port, especially for small vessels that can fish very close to land, therefore different approaches are needed to identify when fishing activities are occurring. During a fishing trip, vessels' movement profiles depend on the activity they are engaged with. Steaming to fishing grounds is usually associated with higher speeds and a relatively straight trajectory, while fishing activities are generally conducted at lower speeds and with greater sinuosity. Consequently, methods to identify different activities during a trip generally use either speed, or speed and turning angle (a measure of sinuosity) to identify fishing. Thus far, the main methods used include: (*i*) an overall speed threshold, inferred from a sample of vessel movements and known activities or estimated from expert knowledge [16–19]; (*ii*) Gaussian mixture models (GMM) fitted using an expectation-maximization (EM) algorithm [20], and; (*iii*) hidden Markov models (HMMs) [21–24]. The EM algorithm implements an iterative procedure that uses maximum-likelihood estimates to cluster the data assuming that they come from a mixture of normal distributions [25] but without assuming a temporal structure. By contrast, HMMs are doubly stochastic models that specifically account for temporal dependence in the data generating mechanism. In the classic HMM formulation, each observation (e.g. speed and turning angle) is assumed to be generated by an underlying, unobserved state process (fishing activity, in this case). This process is Markovian, so that the probability of a given state at the current time step depends on the state at the previous time step, and observations are assumed to be independent, conditional on the underlying state [26]. The nature of temporal dependence is inferred through the transition probability matrix, which is estimated during model fitting and provides estimates of the probability of transitioning between states from one time step to the next, optionally as a function of time-varying covariates. HMMs have gained popularity in the animal movement ecology literature as a method for classifying movement behaviour (e.g. [27,28]). Modelling the movement of fishing vessels and associated fishing activities is very similar. Characteristic distributions of vessel speed and turning angle are commonly associated with hidden states, which can be thought of as corresponding to fishing activities (e.g. higher speed and small turning angle corresponding to steaming), and in our case, validated by records from on-board observers.

Increasingly, tracking devices are being trialled in SSF across the world. In the Republic of Congo, small scale fishing vessels were equipped with low-cost GPS trackers to give insights into the behavioural dynamics of the fleets and the location of fishing grounds [29]. For some fleets in Europe, it is now

mandatory to install a positional tracker to be allowed to fish, for example, in Scotland, the electro-fishery for razor clam or the dredge bivalve fishery in Denmark [30]. There are also several pilot trials assessing the feasibility of collecting positional data for SSF [30]. Nevertheless, these experiences are relatively recent and up-to-date methods for identifying different activities during a trip have been used almost exclusively to analyse data from LSF, even though SSF constitute the largest proportion of fishing vessels globally. For example, in almost all European states, SFF represent over 80% of the total fleet [31]. In addition, SSF require higher reporting frequencies to identify fishing activities, e.g. 60 s in vessels using pots and targeting lobsters and crabs in Scotland [32], and 5 min reporting interval in the razor clam dredge fishery in Ireland [9] compared to 30–120 min reporting from VMS units used for LSF (i.e. [33–35]). Compulsory reporting of positional information in SSF together with the anticipated frequency of reporting [32] will rapidly generate large amounts of data on a daily basis, some of which may need near real-time analyses for compliance purposes. It is therefore essential to identify statistical approaches that will facilitate computationally efficient analyses, while maintaining acceptable levels of accuracy and minimizing potential bias.

In this study, we aim to infer activities of fishing vessels using pots and traps from global navigation satellite system (GNSS) data collected by on-board observers, while prioritizing computational efficiency. To do this, we compare five approaches with varying levels of complexity: a single overall GMM applied to speed values from all vessels; a trip-based GMM capable of assigning a speed threshold value for each vessel; a trip-based binary clustering GMM using speed and turning angle; a univariate HMM using only speed; and a multivariate HMM including speed and turning angle. We judge the performance of each approach by comparing its outputs to observers' ground-truthed data, and by its computational efficiency (model run time). The ground-truthed data came from 115 fishing trips, conducted by vessels using static gear (pots and traps) and targeting lobsters and crabs. We compare the spatial distribution of the fishery estimated by the best approach with the one estimated from these observations. We recommend a pragmatic approach when selecting the most appropriate method for inferring the spatial distribution of fishing activity based on vessel trajectory parameters; one which balances accuracy with efficiency.

# 2. Material and methods

## 2.1. The fishery

This study focuses on small scale fishing vessels (defined in Scotland as 12 m or less, overall vessel length) using static gear (pots or creels) in the East and West coast of Scotland and targeting species such as lobsters (*Homarus gammarus*), crabs (*Cancer pagurus* and *Necora puber*) and prawns (*Nephrops norvegicus*). In 2016, these 'creeler' vessels represented almost 70% of Scotland's small-scale fishing fleet (approx. 1183 of 1734 vessels) and generated around £48.6 million for the Scottish economy in 2016 [36]. Creelers' overall length ranges from 3.6 to 11.9 m, but more than 80% of the vessel were between 5 and 11 m in length in 2016 (Marine Scotland 2016, unpublished data).

On a typical fishing day, after leaving port, fishers steam at relatively high speeds to fishing grounds, where 'strings' or 'fleets' (sets of approx. 10–50 creels per string if targeting lobsters or crabs and approx. 30–100 if targeting prawns) (T. Mendo 2018, personal observations) that have been set usually between 1 and 3 days prior (Marine Scotland, 2017) are marked with surface marker buoys. When the skipper approaches a marker buoy, the speed of the vessel decreases to approximately 1 knot until close enough to hook the buoy and begin the process of hauling creels. The movement of the vessel is then determined by a combination of the hauler (usually a rotating drum) used to retrieve each creel at very low speeds, and the effects of wind and current (tide) acting upon the vessel. The relative influence of these factors will vary depending upon the type, size and configuration of the vessel. However, the vessel is normally positioned down wind or current to ensure that the creel or pot line remains clear of the vessel during hauling. The line of creels or pots usually offers sufficient resistance (drag) to cause the vessel to orientate relative to the creel being hauled—the vessel effectively pivoting on the hauling device. Once on board, the creel is opened, and its contents sorted (crabs, lobsters, or prawns above the minimum legal size retained, and the rest returned to sea). Creels are then quickly re-baited and positioned at the back of the boat in an ordered manner to allow for creels and lines to be deployed at intermediate or higher speeds. Directly prior to deployment, the skipper either travels to a new fishing ground or stays roughly in the same location. The deployment of strings is carried out while the vessel keeps a steady heading. The first marker buoy is deployed which, when drawn taut, drags the first creel off the stern of the vessel, the weight of each sinking creel then pulls the next one off the stern of the vessel resulting in a

roughly straight line of creels. After deployment, the skipper will steam to another fishing location and repeat this process. This cycle is repeated until the fishing trip is completed and the boat returns to port.

## 2.2. Data pre-processing

Data were collected from 115 fishing trips that took place May 2017–July 2018, as part of the Scottish Inshore Fishing Integrated Data System (SIFIDS) project which aims to develop an integrated system for the collection, collation, analysis and interrogation of data from the Scottish inshore fishing fleet. A total of 94 fishing vessels operating in 30 different ports were selected from locations around Scotland. Ports were selected based on the number of annual trips conducted, and then discussions with fishery officers at Marine Scotland to ensure higher skipper participation rates and coverage of both the East and the West Coast. For logistical reasons, the north of Scotland (including Orkney and Shetland Islands) were excluded from the survey design (see electronic supplementary material, S1 and figure S1.1 for map of locations, exact location not shown, due to confidentiality agreements with fishers). For each trip, GNSS data were collected by an on-board observer using a handheld Garmin Etrex 20 where GNSS positions were recorded at 1 s intervals. Observers also recorded the registry of shipping (RSS number), target species, departure time and several vessel activities (time of hauling events, time of re-deployment of creels and time when the vessel reached port at the end of each trip).

## 2.3. Methods used to identify hauling activities

Activities recorded by on-board observers were matched with the GNSS tracking data using the time in which each activity occurred. A 60 s polling (sampling) frequency was deemed a good interval to identify fishing activities in this fleet by Mendo *et al.* [32]. Our knowledge of the fishery indicates that there are three main types of behaviours that occur during a fishing trip: (1) steaming to, from and in between fishing grounds; (2) deploying gear (also called shooting) and; (3) hauling gear. Exploratory data analysis of the distribution of vessel speeds (in knots) during each behaviour, as recorded by the on-board observers, suggested that speeds at deployment overlapped speeds during steaming in some vessels (electronic supplementary material, S1 and figure S1.2), sometimes making them indistinct from each other. However, estimating hauling appropriately is sufficient to estimate the spatial distribution of the fishery (creels can only be hauled where they have been deployed in the first instance). For the sake of completeness, we fit models with three states to all of the data, even if these three states were not distinguishable in all vessels. However, to make the comparison fairer and allow for the lack of a distinct third state in some vessels, we evaluate the performance of our methods only by comparing model outputs for hauling or 'not hauling', where 'not hauling' includes both deploying and steaming behaviour. All analyses were conducted on a desktop computer (Intel® Core™ i7–5820 K @3.30 GHz with 32 Gb RAM x64-bit Windows 10 Pro OS). It is common practice to use step length (distance travelled between locations) instead of speed in hidden Markov models, so we converted speed in knots to distance travelled in metres for each 60 s interval (sampling interval for positional data) for those models, which is essentially speed in metres per minute. We call both measures of distance covered per time unit (knots and m min$^{-1}$) speed throughout this section.

The five methods used to identify fishing activities are described below. We include the code required to implement these methods (electronic supplementary material, S2) together with an example dataset [37].

### 2.3.1. 'Overall' speed GMM

Speed data for all trips were combined, and EM was used to estimate the parameters of the multimodal speed distribution within a univariate GMM. We implemented this model using the mixtools package [25] in R [38]. Three underlying univariate normal distributions were assumed to correspond to hauling, deploying gear and steaming ($k = 3$). Even though in some vessels, the distribution of vessel speeds overlapped between steaming and deployment, we cannot anticipate which vessels will show two (1 = hauling, 2 = combined steaming and deployment) rather than three distributions (1 = hauling, 2 = deployment and 3 = steaming). Therefore, since we know there to be three true underlying states, we chose to use three underlying behaviours for all vessels in order for the model to be applicable to the whole fleet. Starting values for the mean and the standard deviations for each underlying distribution were estimated visually using a histogram showing the multimodal distribution of speed (electronic supplementary material, S1 and figure S1.3).

To define the upper limit to the distribution for hauling, we used the estimated mean for hauling and added two standard deviations to it. All positional records that had a speed greater than this were labelled as 'not hauling'.

### 2.3.2. Trip-based GMM with speed only

Similar to the approach in 2.3.1., we used EM to estimate the parameters of the GMMs fitted to the speed frequency distribution from each trip. Means and standard deviations for the component normal distributions estimated in 2.3.1. (above) were used here as starting values for the model fitted to each fishing trip. Using a normal distribution for a strictly positive-valued measure, such as speed, might result in unrealistic, negative speed values predicted for hauling activities. However, the means of the hauling speed distributions were not close enough to zero to result in negative mean estimates. In addition, we were not interested in the lower tail of the distribution of hauling, only its intersection with steaming, which lay in the upper tail of the distribution of hauling speed. Therefore, the upper threshold for hauling for both 2.3.1. (overall speed) and 2.3.2. (trip-based speed) was calculated as the estimated mean of the hauling speed distribution plus two times the estimated standard deviation. As in 2.3.1. (above) all positional records that had a speed greater than this were labelled as 'not hauling'.

### 2.3.3. Trip-based GMM with speed and turning angle

The EM binary clustering (EMbC) algorithm was used to carry out maximum-likelihood estimation of the parameters of a bivariate GMM [39]. This unsupervised classification approach uses speed and angle, and clusters the observations into high/low speed and high/low angles using likelihood expectation maximization producing four possible combinations: high speed/large turning angle, high speed/small turning angle, low speed/large turning angle and low speed/small turning angle [25,38].

### 2.3.4. Trip-based HMM with speed only

The R package moveHMM [28] was used to classify the step length of each vessel during a fishing trip into three underlying states (steaming, hauling and deploying). In moveHMM models are fitted via numerical optimization of the likelihood, which requires setting initial values for the model parameters (e.g. mean and standard deviation for each distribution, [28]). Due to the great variability in vessel overall length and engine power in the fleet, starting values for the mean and standard deviation of each underlying state distribution for step length were estimated using a GMM as in 2.3.2. (above) after converting knots to metres per 60 s interval. The numerical maximization routine to identify the global maximum failed to converge for all trips using the same set of starting values; therefore, different sets of initial values of the speed distribution were explored. Out of a wide range of starting values, the only configuration that yielded a fit for all vessels was to provide a small standard deviation for hauling, and a big standard deviation for steaming. This was informed by the spread evident in the observer records for each distribution. In contrast to the GMM approach used in 2.3.1. and 2.3.2., the positive-valued gamma distribution was used here to model speeds.

### 2.3.5. Trip-based HMM with speed and turning angle

This approach is the same as in 2.3.4. except for the inclusion of turning angle as a second state variable, which we modelled using the wrapped Cauchy distribution. Starting values for the mean and standard deviation of each underlying state distribution for speed were estimated as explained in 2.3.4. We estimated the concentration of the turning angle distributions as well as the mean, and provided starting values to reflect straight, directed movement during steaming and undirected, more sinuous movement during hauling.

## 2.4. Performance assessment

To assess the performance of each method, we compared the accuracy of its output to observers' ground-truthed data on hauling activities and measured its computation time. We estimated *accuracy*, defined as the number of correctly classified instances (for both hauling and not hauling) with respect to their total number of locations. The error rate per trip was calculated by dividing the total number of incorrectly assigned positional records by the total number of positional records in each trip. This was then multiplied by 100. The true-positive rate measures the percentage of actual hauling events that were correctly identified as such. The false-positive rate measures the percentage of the non-hauling events wrongly identified as hauling out of the total number of actual non-hauling events. The time elapsed for each analysis was estimated using the base R function system.time() [38].

## 2.5. Reasons for misclassification under the 'best' method

We considered the 'best' method to be the one that correctly identified hauling activities most often and had the shortest computation time. To explore the reasons for misclassification in the selected method we plotted correctly identified positional records, false positives (those which are categorized as hauling but are non-hauling) and false negatives (those classified as non-hauling when in reality they are hauling) in space during a fishing trip. We used observer records to identify the main reasons for misclassifications. From visual inspection, it was clear that most errors occurred just before or after a hauling event. We used binomial generalized linear models to relate the proportion of positional records correctly identified to 5–10 min time bins before, during and after a hauling event, respectively. *Post hoc* multiple comparisons were evaluated using a Tukey test with a Holms correction from the multcomp R package [40].

The effect of vessel size class and species targeted (crabs, lobsters or prawns) on the per-trip error rates was evaluated using linear multiple regression with an interaction term between vessel length and target species. Normality and homogeneity of variances were assessed using a Shapiro–Wilks and Bartlett test, respectively.

## 2.6. Comparing the spatial distribution of hauling activities: known versus modelled

Once we selected our approach, the resulting spatial distribution was compared to the spatial distribution of hauling activities known from observations. Each haul was defined by the retrieval of the start and end buoy of a set of creels, and as times were recorded by on-board observers for each of these, they could be combined with the GNSS tracking data to identify hauling events in space. Consecutive records identified as hauling were considered a unique hauling event.

Once hauling events had been identified, the area covered during each event was estimated by joining all positional records from that haul and adding a buffer radius of 50 m around these positions as a proxy of effective area fished for each haul. This radius was used to include all locations from which individual target species could have travelled to the creels, prior to capture. This radius was informed by different estimates of the effective fishing area of creels (e.g. 43–57 m [41] and 21–34 m [42] for *Cancer pagurus*, and 2–12 m [43] and 28.8 m [44] for *Homarus americanus*. No studies could be found for *H. gammarus* or *Nephrops norvegicus*).

The area for each hauling event was calculated, summed and compared to the total effective area fished (estimated with the observer's data). The difference between the area resulting from our approach and the real area fished (false positives, overestimating area) and the difference between the real area fished and the estimated area (false negatives, underestimation of area fished) were calculated using the gIntersection function, from the rgeos R package [45].

# 3. Results

In total, 2 886 110 GNSS records were downloaded from GNSS devices during the study period of 15 months. These records were used to define specific trips (e.g. figure 1) by assigning each Trip ID, a trip starting time and end time as recorded by observers (2 783 779 records remaining). Duplicate records (3040 records) were deleted. No erroneous positions on land were identified. Spatial buffer zones of 200 m were set around landing ports to avoid incorporating locations with low speeds as a result of transiting harbour areas (2 777 489 records remaining). This conservative threshold was based on observations of vessels deploying gear at distances less than 300 m from port (T. Mendo 2018, personal observation). These positional records were subsequently subsampled to a 60 s polling interval, see [32]. The final number of GNSS records used in the analysis was 46 277.

## 3.1. Classification of positional records

All five approaches performed well in identifying fishing activities. The best overall accuracy (proportion of correctly classified instances) was similar across approaches (table 1) but was highest for the trip-based GMM and lowest for the GMM using an overall speed threshold. The maximum error rate per trip was obtained for the trip-based HMM with step length and turning angle, followed by the trip-based GMM with speed and turning angle, and lowest for the HMM with speed only and the trip-based GMM with speed only. True-positive rates ranged from 94.2% with the overall speed threshold to 97.5% with the trip-based HMM with speed only. The lowest false-positive rate was obtained with the trip-based

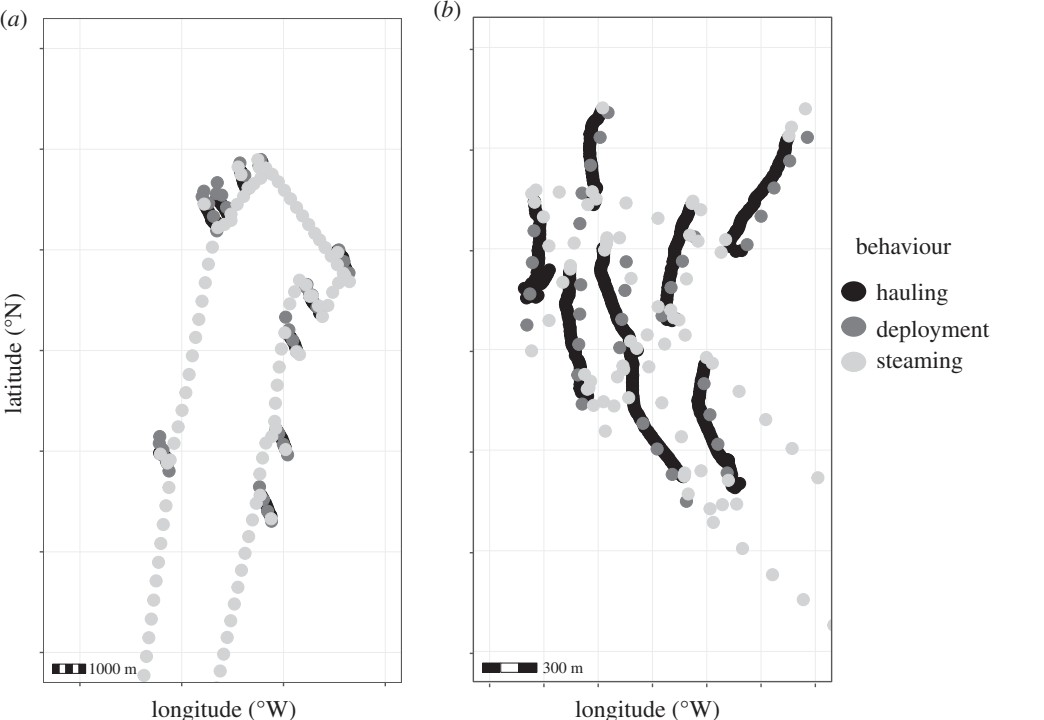

**Figure 1.** Examples of segments of individual fishing trips (*a,b*) showing positional records (black = hauling, dark grey = deploying, light grey = steaming) every 60 s. Coastline and coordinates not shown to maintain vessel anonymity.

**Table 1.** Accuracy, per trip error rate (%), false-positive rate, false-negative rate and time elapsed for computation of 115 trips using five different approaches.

| | overall speed | trip-based | | | |
| --- | --- | --- | --- | --- | --- |
| | GMM speed | GMM speed | GMM speed and angle | HMM speed | HMM speed and angle |
| accuracy (%) | 91.14 | 92.3 | 91.37 | 91.97 | 91.68 |
| per trip error rate (% range) | (2.16–33.33) | (2.05–28.95) | (1.57–33.7) | (1.81–25.92) | (1.80–35.94) |
| true-positive rate | 94.23 | 95.69 | 97.37 | 97.58 | 97.19 |
| false-positive rate | 11.28 | 10.34 | 13.23 | 12.37 | 11.95 |
| time elapsed (seconds) | 1.56 | 3.06 | 15.48 | 197.09 | 352.20 |

speed-only GMM. There was a vast difference (up to 100 times) between the time elapsed to run any of the GMMs (less than 20 s) for 115 trips and the time elapsed to run HMMs (200-350 s) (table 1). Based on these performance statistics, and the longer time required to run HMMs, the trip-based speed-only GMM was considered to achieve most satisfactory results in terms of an overall lower accuracy rate, lower maximum error rate per trip, lower false-positive rate and computing time.

## 3.2. Reasons for misclassification using the trip-based GMM with speed only

Visualization of trips showed that the false positives and false negatives related to hauling were mainly located just before a hauling event (figure 2, left panel: before hauling). The proportion of positional records correctly identified was lowest in the 10 min before hauling, gradually increasing with time since the hauling event (figure 3). The proportion of positional records correctly identified was lowest after 25 min of hauling activities. After a hauling event, there was no clear trend in the proportion of positional records correctly identified. False positives in classifying hauling activities occurred when fishers would repair ropes or creels, or band lobsters just after a hauling event, when they were fishing

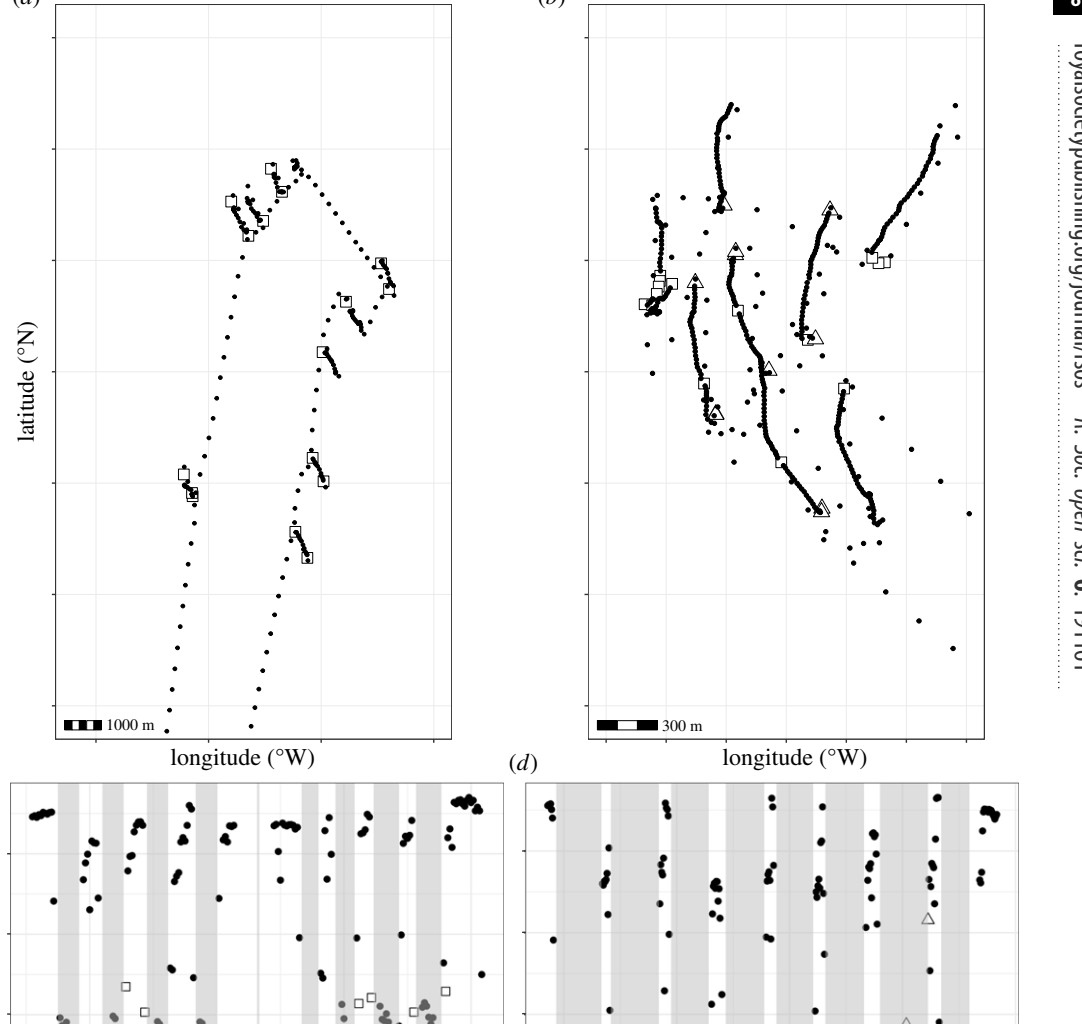

**Figure 2.** Examples of classification results for the trip-based GMM, showing individual fishing trips in space (*a,b*) and corresponding vessel speed during the day (*c,d*). Each symbol represents a positional record. Grey bands in c and d are the times when hauling was observed. False-positive records (squares) are those which fall outside the hauling event, but are categorized as hauling. False negatives are those records which fall inside a hauling band but are classified as non-hauling (white triangles). Coastline and coordinates not shown to maintain vessel anonymity.

for bait (usually mackerel caught using hand lines) or when they would stop activities for tea or lunch (electronic supplementary material, S1 and figure S1.4). The relationship between vessel length and error rate varied depending on the main species targeted ($F = 4.596$, d.f. = 5,103, $p < 0.001$). In the case of crabs and lobsters, there was a decline in error rate as vessel length increased, while for *Nephrops*, the error rate did not increase with vessel size (electronic supplementary material, S1 and figure S1.5).

## 3.3. Spatial footprint of hauling activities: observed versus estimated

The fishing area covered during 115 fishing trips was 63.21 km². The trip-based speed-only GMM resulted in an estimate of 68.12 km² for the total fishing area. The estimated spatial extent correctly captured 97% of the true spatial extent of active fishing (3.04% underestimation of the true area fished, i.e. figure 4, false negatives). An additional 10% of the true spatial extent was falsely identified as part of the fishing area; however, these areas were mostly located near real fished areas (i.e. figure 4, false positives).

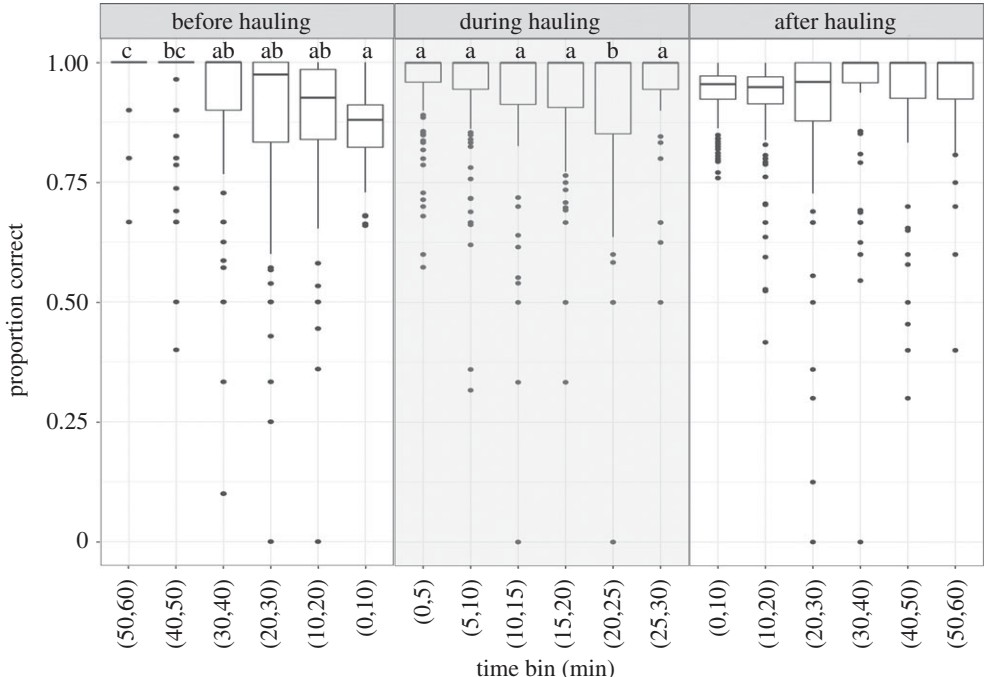

**Figure 3.** Proportion of positional records from observer trips that were correctly identified by the trip-based speed-only GMM, where 1 means all positional records correctly identified as hauling, and 0 means no trips correctly identified as hauling. These proportions are plotted against time (min) for three hauling periods: before hauling (between 60 min before hauling and the beginning of hauling), during hauling (while hauling is taking place—lasting up to 30 min) and after hauling (between the end of hauling and 60 min after hauling ended). The boxes show the interquartile range, the bold lines inside the boxes show the median proportion, the bars show the 95% CI, and the points show the outliers. Differences in letters above the boxplots indicate significant differences among time bins (Tukey's test, $p < 0.05$). Time bins that share letters e.g. ab and a, are not significantly different at the 5% level.

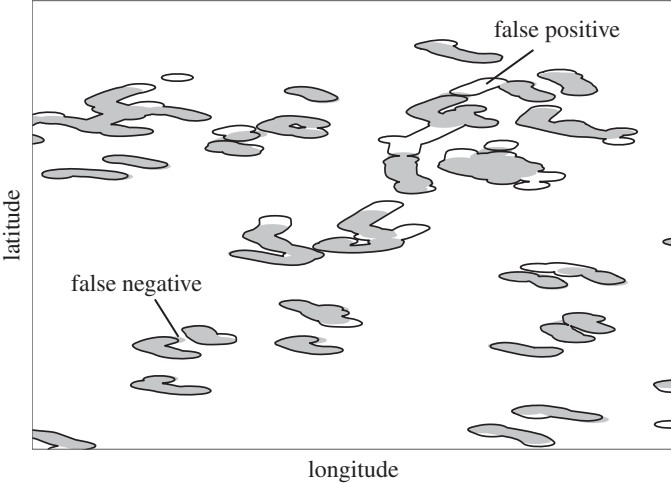

**Figure 4.** Map showing a subset of areas fished identified by observers (grey areas) and identified by the trip-based GMM (black contour lines), showing false positives (white areas inside a black line) and false negatives (grey areas outside the black contour line). Coordinates not shown due to confidentiality agreements with fishers.

## 4. Discussion

Collection of spatial information from SSF fleets at nationwide levels will pose new challenges associated with the large amounts of data that will be collected. This study investigates the performance of existing approaches used to predict fishing activities (GMMs and HMMs), while taking into consideration misclassification of non-hauling activities as hauling. The five different approaches examined in this

study performed similarly in identifying fishing activities in space albeit with significant computation time differences. In the case of the Scottish SSF fleet, we found that a GMM using speed only, and applied to each vessel trip separately, achieved the best compromise of accuracy, an overall lower maximum error rate per trip, a lower false-positive rate and computational speed. These results are well placed to inform the data analysis protocols that will need to be developed for long-term monitoring of fishing areas by SSF fleets.

The trip-based speed-only GMM performed well in predicting fishing (hauling) activities without the computational burden of HMMs. Furthermore, it yields a similar degree of accuracy to trip-based HMMs without the need for multiple restarts or initial re-parametrization. Initial values provided to the EM algorithm (mean and standard deviation for each state) for estimating the parameters of the GMM were easily estimated by visual inspection of the multimodal distribution of speeds. In comparison, HMMs required many attempts (testing different standard deviations for the step length distribution for each state) to overcome convergence issues that can arise when starting values are poorly chosen [28]. In reality, the convergence problems for HMMs in this application likely arise from forcing all trips to have three functional states, based on our knowledge of the case study; meanwhile, in some trips, there were only two movement states. From a practical point of view, not having to pay close attention to starting values would again favour the use of the trip-based GMM for speed using an EM algorithm for estimation.

HMMs are powerful models for time series and provide additional inferences about the evolution of a process through time, compared to GMMs. When these quantities are of interest the additional computation time is justified. However, at this stage in the development of using positional data from SSFs the main interest is in the rapid classification of fishing behaviour, which, based on our results, is better addressed using an approach which does not explicitly account for the serial dependence in the data. HMMs will become relevant to the analysis of SSF data in the future if interest shifts to the switching behaviour between behaviours, or the duration of behaviours, which might suggest something about the distribution and abundance of target prey.

The trip-based speed-only GMM took approximately 3 s to classify positional records from 115 trips (46 277 records) into hauling and not hauling activities (approx. 0.026 s per trip). Assuming these 115 trips are representative of the fleet and that all vessels would carry a device to record positional data at a 60 s polling interval, classifying fishing activities in the circa 1540 SSF vessels operating in Scotland would take approximately 40 s on a day in which all vessels would be engaged in fishing activities. If we take into consideration the total number of fishing trips conducted annually in Scotland (111 909 trips in 2016, Marine Scotland, 2016, unpublished data), the classification of all annual trips would result in 48 min of computing time. These results highlight the feasibility of designing a monitoring system that could efficiently generate information on main fishing grounds, fishing effort, or monitoring of compliance to regulations for the Scottish SSF fleet. This information opens up the potential to explore different fisheries management regimes to improve compliance, efficiency and sustainability. Rights-based and co-management systems, which are an increasing feature in SFF, must be underpinned by data of known provenance, quality and coverage. Fishers, those that regulate their activities and other key stakeholders, must have common access to these data in ways that allow the fishery to be monitored and managed effectively without compromising commercial or operational sensitivities. An important feature of securing both compliance and effective fisheries management is to engage fishers in data collection. Combining robust and secure spatio-temporal data collection with catch and bycatch recording using other mobile technologies can form the basis of a participatory fisheries management regime. In summary, our ability to collect data cost-effectively from SSF globally has profound implications for fisheries management. To realize this potential, we will need to develop new systems and processes for analysing and interpreting the rapidly increasing volumes of data that will be produced. The work presented here can be regarded as a proof-of-concept for one such analytical approach.

Initiatives to make positional data recording a statutory requirement for the whole SSF fleet (e.g. [13] in England) should consider extending this analysis to other gears, in order to identify the most suitable approach applicable to each fishery. While for vessels using pots and traps in Scotland all five approaches performed similarly (except in computational time), other fisheries might benefit from using HMMs over mixture models, especially by adding turning angle information. In this study, the distribution of turning angles was very similar in all three behaviours (steaming, deploying and hauling). Indeed, during hauling, vessels would haul strings (sets of creels) in a fairly straight course (e.g. in figure 1). Therefore, the turning angle distribution during hauling was not distinct from the distributions during steaming or deploying. This suggests that fitting an HMM where we expect the state-dependent distribution of turning angles in the slower, less directed state to be less concentrated, constitutes a model misspecification in this case, where the slower behaviour is in fact also directed. Additionally, the Markov property, whereby the future state depends on the present state, might be slowing down

the switch between hauling and not hauling. In this fishery, vessels switch rapidly between hauling and not hauling and vice versa. In other fisheries, marked differences in movement patters during fishing activities might favour the use of HMMs to increase the performance of the identification of fishing activities (e.g. hidden semi-Markov models performed best in the purse seine Peruvian anchovy fishery [23]). This analysis would have to be extended or revisited before it could be used with data gathered in LSF (from VMS or AIS) as dissimilarities in reporting frequencies might result in different computational times and accuracy in detecting fishing activities.

# 5. Conclusion

Our assessment of five statistical approaches for classifying positional records from SSF vessels points towards the need for a case-by-case treatment. We have shown that all five approaches performed well in identifying fishing behaviour; however, the best-suited method among the ones we present will vary depending on the use case. In the case of the Scottish inshore SSF, a trip-based GMM was able to accurately and efficiently identify fishing activities based on the separation of the speed distributions of fishing behaviour. With the positional records accurately classified, we show that the estimated spatial extent of fishing activities was approximately 92% of the true area fished (as reported by on-board observers). These results highlight the feasibility of designing a monitoring system that could efficiently generate information on core fishing grounds, fishing intensity, or monitoring compliance to regulations, for a nationwide SSF fleet. These explicitly spatial results produced from anonymized vessel behaviour, facilitate their direct inclusion in marine spatial planning activities without disclosing sensitive information. Finally, inferring the spatial location and extent of active fishing will inform which areas may need conservation management in the future, depending on the impact that SSF fishing activities might have on the seabed.

Ethics. We were given permission to use the data by its custodian, Marine Scotland, and we perceive no ethical issues due to anonymity of the fishing vessels involved and concealment of true locations.

Data accessibility. Example data on fishing vessel locations are available within the Dryad Digital Repository: https://dx.doi.org/10.5061/dryad.k80bp46 [37].

Authors' contributions. T.M., M.J. and S.S. conceived of the study and interpreted the results. T.M. carried out the analysis and wrote the draft manuscript. M.J. and S.S. also contributed to discussion and later iterations of manuscript text. T.P. revised the manuscript critically, contributed to the code and the methodology and interpreted the results. All authors contributed to editing the manuscript, and all authors agree to be accountable for all aspects of the work and gave final approval for it to be published.

Competing interests. We have no competing interests.

Funding. This study (T.M., M.J. and S.S.) was funded by the European Maritime Fisheries Fund 'Scottish Inshore Fisheries Integrated Data System' (grant reference no. SCO1434). T.P. was supported by a Newton International Fellowship, funded by the Royal Society (grant reference no. NF170682).

Acknowledgements. This research would not have been possible without the support of the fishers who provided access to their vessels and data, the project facilitators: Kyla Orr, Ali McKnight and Kathryn Logan and the on-board observers: Guy Pasco, Grant Course and Ash Royton. Thanks to Jim Watson, Nick Lake, Nick Jones, Debbie Russell and Matt Carter for helpful discussions, and to all district Fishery officers that contributed to the identification of ports to conduct research. John Thompson and Hannah-Ladd Jones were responsible for managing the Scottish Inshore Fisheries Integrated Data System (SIFIDS) project from which this research is an output. The authors are very grateful to Jennifer Pohle for providing invaluable technical input to the methodology, which greatly improved the paper.

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
