## [Reviewer comments · Royal Society Open Science]

Review History

RSOS-191161.R0 (Original submission)

Review form: Reviewer 1 (Tommaso Russo)

Is the manuscript scientifically sound in its present form?

Yes

Are the interpretations and conclusions justified by the results?

Yes

Is the language acceptable?

Yes

Do you have any ethical concerns with this paper?

No

Have you any concerns about statistical analyses in this paper?

No

Recommendation?

Accept with minor revision (please list in comments)

Comments to the Author(s)

The manuscript is very well written and clear. The results are sound and the discussion and conclusions are well supported and coherent.

My only suggestion to the authors is that, maybe, it could be important to conjecture about the potential difference (both in terms of accuracy and computational time) of these methods for VMS/AIS data for LS vessels. This could also represent a link to future studies.

Review form: Reviewer 2**Is the manuscript scientifically sound in its present form?**

Yes

Are the interpretations and conclusions justified by the results?

Yes

Is the language acceptable?

Yes

Do you have any ethical concerns with this paper?

No

Have you any concerns about statistical analyses in this paper?

No

Recommendation?

Accept with minor revision (please list in comments)

Comments to the Author(s)

Objectives are well identified, methodological approach is correctly set out, conclusions are essentially coherent. But, of course, the paper can be improved. Obviously, the underlying problem (SSF positional data collected and validated by on-board observers) would deserve two additional extensions:

i) A more detailed introduction on other previous experiences on SSF positional data collection and processing (if there are). Bibliographic references mostly refer to LSF and not SSF experiences.

ii) A more in-depth analysis about regulation systems in SSF and the mechanisms for achieve compliance and efficiency (i.e. rights-based systems or co-management systems). It is a key point to design a feasible monitoring system for a country-wide SSF.

A reflection in any of these directions could enrich the work.

The only fundamental observation I should like to make is that the paper is strictly related to Scottish vessels using pots and traps and so it is not replicable/applicable "as is" to other gears/fisheries. I see, however, that the authors recognize these conditions and they prudently place their assessments and conclusions.

So, in my opinion, the article is admissible with minor revisions (i) but the authors have the opportunity to complete and improve it (ii)

Decision letter (RSOS-191161.R0)

13-Aug-2019

Dear Dr Photopoulou

On behalf of the Editors, I am pleased to inform you that your Manuscript RSOS-191161 entitled "Identifying fishing grounds from vessel tracks: model-based inference for small scale fisheries" has been accepted for publication in Royal Society Open Science subject to minor revision in accordance with the referee suggestions. Please find the referees' comments at the end of this email.

The reviewers and handling editors have recommended publication, but also suggest some minor revisions to your manuscript. Therefore, I invite you to respond to the comments and revise your manuscript.

- Ethics statement

- Data accessibility

If you wish to submit your supporting data or code to Dryad (<http://datadryad.org/>), or modify your current submission to dryad, please use the following link:
<http://datadryad.org/submit?journalID=RSOS&manu=RSOS-191161>

- Competing interests

- Authors' contributions

AB carried out the molecular lab work, participated in data analysis, carried out sequence alignments, participated in the design of the study and drafted the manuscript; CD carried out

the statistical analyses; EF collected field data; GH conceived of the study, designed the study, coordinated the study and helped draft the manuscript. All authors gave final approval for publication.

- Acknowledgements

- Funding statement

Because the schedule for publication is very tight, it is a condition of publication that you submit the revised version of your manuscript before 22-Aug-2019. Please note that the revision deadline will expire at 00.00am on this date. If you do not think you will be able to meet this date please let me know immediately.

- 1) A text file of the manuscript (tex, txt, rtf, docx or doc), references, tables (including captions) and figure captions. Do not upload a PDF as your "Main Document";
- 2) A separate electronic file of each figure (EPS or print-quality PDF preferred (either format should be produced directly from original creation package), or original software format);
- 3) Included a 100 word media summary of your paper when requested at submission. Please ensure you have entered correct contact details (email, institution and telephone) in your user account;
- 4) Included the raw data to support the claims made in your paper. You can either include your data as electronic supplementary material or upload to a repository and include the relevant doi within your manuscript. Make sure it is clear in your data accessibility statement how the data can be accessed;

5) All supplementary materials accompanying an accepted article will be treated as in their final form. Note that the Royal Society will neither edit nor typeset supplementary material and it will be hosted as provided. Please ensure that the supplementary material includes the paper details where possible (authors, article title, journal name).

on behalf of Prof Kevin Padian (Subject Editor)
openscience@royalsociety.org

Reviewer comments to Author:
Reviewer: 1

Comments to the Author(s)

The manuscript is very well written and clear. The results are sound and the discussion and conclusions are well supported and coherent.

My only suggestion to the authors is that, maybe, it could be important to conjecture about the potential difference (both in terms of accuracy and computational time) of these methods for VMS/ AIS data for LS vessels. This could also represent a link to future studies.

Reviewer: 2

Comments to the Author(s)

Objectives are well identified, methodological approach is correctly set out, conclusions are essentially coherent. But, of course, the paper can be improved. Obviously, the underlying problem (SSF positional data collected and validated by on-board observers) would deserve two additional extensions:

i) A more detailed introduction on other previous experiences on SSF positional data collection and processing (if there are). Bibliographic references mostly refer to LSF and not SSF experiences.

ii) A more in-depth analysis about regulation systems in SSF and the mechanisms for achieve compliance and efficiency (i.e. rights-based systems or co-management systems). It is a key point to design a feasible monitoring system for a country-wide SSF.

A reflection in any of these directions could enrich the work.

The only fundamental observation I should like to make is that the paper is strictly related to Scottish vessels using pots and traps and so it is not replicable/applicable "as is" to other gears/fisheries. I see, however, that the authors recognize these conditions and they prudently place their assessments and conclusions.

So, in my opinion, the article is admissible with minor revisions (i) but the authors have the opportunity to complete and improve it (ii)

Author's Response to Decision Letter for (RSOS-191161.R0)

See Appendix A.

Decision letter (RSOS-191161.R1)

04-Sep-2019

Dear Dr Photopoulou,

I am pleased to inform you that your manuscript entitled "Identifying fishing grounds from vessel tracks: model-based inference for small scale fisheries" is now accepted for publication in Royal Society Open Science.

Royal Society Open Science operates under a continuous publication model (<http://bit.ly/cpFAQ>). Your article will be published straight into the next open issue and this

will be the final version of the paper. As such, it can be cited immediately by other researchers. As the issue version of your paper will be the only version to be published I would advise you to check your proofs thoroughly as changes cannot be made once the paper is published.

on behalf of Mr Andrew Dunn (Associate Editor) and Kevin Padian (Subject Editor)
openscience@royalsociety.org

Associate Editor Comments to Author (Mr Andrew Dunn):

Associate Editor: 1

Comments to the Author:

(There are no comments.)

Reviewer comments to Author:

Appendix A

Dr Theoni Photopoulou
School of Biology
University of St Andrews
St Andrews, Scotland

22 August 2019

Dear Prof Kevin Padian,
Academic Editor, Royal Society Open Science

REVISED SUBMISSION OF MANUSCRIPT RSOS-191161
Identifying fishing grounds from vessel tracks: model-based inference for small scale
fisheries

Many thanks to the two anonymous reviewers for constructive comments on our manuscript submitted to Royal Society Open Science.

Both reviewers make suggestions for additions or improvements to the text. We have followed their suggestions and added details in response to their comments. The changes made can be seen in the table below where we have addressed the reviewers' comments point-by-point. We underline actions taken in response to each comment.

We feel we have addressed the reviewers' comments and hope you will find our responses satisfactory. We look forward to hearing from you and hope that you find our revised manuscript suitable for publication in Royal Society Open Science.

Sincerely,
Theoni Photopoulou, on behalf of all the authors

Responses to Reviewer 1

Comment	Response
My only suggestion to the authors is that, maybe, it could be important to conjecture about the potential difference (both in terms of accuracy and computational time) of these methods for VMS/AIS data for LS vessels. This could also represent a link to future studies.	Thank you for the suggestion. We agree and we have added a sentence at the end of the discussion: This analysis would have to be extended or revisited before it could be used with data gathered in LSF (from VMS or AIS) as dissimilarities in reporting frequencies might result in different computational times and accuracy in detecting fishing activities.

Responses to Reviewer 2

Comment	Response
i) A more detailed introduction on other previous experiences on SSF positional data collection and processing (if there are). Bibliographic references mostly refer to LSF and not SSF experiences.	Thanks for the comment. We have added detail to the fifth paragraph of the introduction regarding other SSF positional data collection and processing: Increasingly, tracking devices are being trialled in SSF. In the Republic of Congo, small scale fishing vessels were equipped with low-cost GPS trackers to give insights to the behavioural dynamics of the fleets and the location of fishing grounds [29]. For some fleets in Europe it is now mandatory to install a positional tracker to be allowed to fish, for example, in Scotland, the electro-fishery for razor clam or the dredge bivalve fishery in Denmark [30]. There are also several pilot trials assessing the feasibility of collecting positional data for SSF [30]. Nevertheless, these experiences are relatively recent and up to date methods for identifying different activities during a trip have been used almost exclusively to analyse data from LSF, even though SSF constitute the largest proportion of fishing vessels globally.
ii) A more in-depth analysis about regulation systems in SSF and the mechanisms for achieve compliance and efficiency (i.e. rights-based systems or co-management systems). It is a key point to design a feasible monitoring system for a country-wide SSF.	Thank you for this suggestion. We have added text to the fourth paragraph of the discussion: This information opens up the potential to explore different fisheries management regimes to improve compliance, efficiency and sustainability. Rights-based and co-management systems, which are an increasing

feature in SSF, must be underpinned by data of known provenance, quality and coverage. Fishers, those that regulate their activities and other key stakeholders, must have common access to these data in ways that allow the fishery to be monitored and managed effectively without compromising commercial or operational sensitivities. An important feature of securing both compliance and effective fisheries management is to engage fishers in data collection. Combining robust and secure spatiotemporal data collection with catch and bycatch recording using other mobile technologies can form the basis of a participatory fisheries management regime. In summary, our ability to collect data cost-effectively from SSF globally, has profound implications for fisheries management. To realise this potential, we will need to develop new systems and processes for analysing and interpreting the rapidly increasing volumes of data that will be produced. The work presented here can be regarded as a proof-of-concept for one such analytical approach.

The only fundamental observation I should like to make is that the paper is strictly related to Scottish vessels using pots and traps and so it is not replicable/applicable "as is" to other gears/fisheries. I see, however, that the authors recognize these conditions and they prudently place their assessments and conclusions.

We have added a sentence in response to reviewer 1's suggestion along these lines, which we hope will also address this comment. Please see the last sentence of the discussion:

This analysis would have been extended or revisited before it could be used with data gathered in LSF (from VMS or AIS) as dissimilarities in reporting frequencies might result in different computational times and accuracy in detecting fishing activities.